# Evaluation of *Streptococcus pneumoniae* as a cause of acute otitis media in Colombia: A prospective study

Wilfrido Coronell-Rodriguez[1], Rosanna Camerano[2], Juan Carlos Alvarado-Gonzalez[1,2,3], Alejandra Puerto[1,2], Josefina Zakzuk[1], Nelson R. Alvis-Zakzuk[2,4], Lina Moyano-Tamara[2], Sebastian Medina[5], Claudia Beltran[5], Maria Betancur[5], Monica Rojas[5], Luis Enrique Farias[5], Hernando Pinzon-Redondo[1,3], Perla Villamor[3], Steven Osorio[1,3,6], Nelson Alvis-Guzmán[2,7] *

1 Universidad de Cartagena, Cartagena, Colombia, 2 ALZAK, Cartagena, Colombia, 3 Hospital Infantil Napoleón Franco Pareja, Cartagena, Colombia, 4 Institución Universitaria Mayor de Cartagena, Cartagena, Colombia, 5 MSD, Bogotá, Colombia, 6 Hospital Serena del Mar, Cartagena, Colombia, 7 Universidad de la Costa, Barranquilla, Colombia

* nalvis@cuc.edu.co

## Abstract

### Objective

Since the introduction of the pneumococcal conjugate vaccine PCV-10 in 2011 its impact on acute otitis media (AOM) in children under five years of age in Colombia was unknown. We aimed to describe the clinical and sociodemographic character-istics of a prospective cohort of patients between 3 and 59 months old attending a children's hospital in Cartagena, Colombia.

### Methods

A prospective cohort study was conducted over a 12-month period from August 5th, 2022 to August 5th, 2023. Diagnosis of AOM was confirmed by an otorhinolaryngolo-gist. Middle ear fluid samples were obtained by swab or tympanocentesis, depending on the presence of spontaneous drainage. Samples with a positive culture for *S. pneumoniae* were sent to the Colombian National Health Institute for serotyping. We also estimated the prevalence of AOM caused by *S. pneumoniae*, the serotype distri-bution and antimicrobial resistance patterns.

### Results

A total of 61 patients were enrolled, 58% were male, the median age was 12 months (IQR: 8–24). The most common isolated microorganisms were *Pseudomonas aeru-ginosa* (14.8%), methicillin-resistant *Staphylococcus aureus* (13.1%), and *Strepto-coccus pneumoniae* (9,8%). Six cases of *S. pneumoniae* were identified, median age was 26.5 months (IQR: 8–45), none had any comorbidities, and only one had a

**Data availability statement:** Data cannot be shared publicly because of sensitive information. Data are available upon request from the Ethics Committee of IMAT Oncomédica (ONC-CEI-CEI-231-2022) (soporteproyectos@alzak.com.co / comitedeeticaeinvestigaciones@gmail.com) for researchers who meet the criteria for access to confidential data.

**Funding:** This study was funded by MSD Colombia, a subsidiary of Merck & Co., Inc., Rahway, NJ, USA. Claudia Beltran, Sebastian Medina, Monica Maria Rojas, Maria Alejandra Betancur and Luis Enrique Farias are MSD employees. This did not interfere with the authors' ability to analyze, interpret the data, or prepare the manuscript.

**Competing interests:** This study was funded by MSD Colombia, a subsidiary of Merck & Co., Inc., Rahway, NJ, USA. Claudia Beltran, Sebastian Medina, Monica Maria Rojas, Maria Alejandra Betancur and Luis Enrique Farias are MSD employees. This does not alter our adherence to PLOS ONE policies on sharing data and materials.

history of previous AOM episodes. Five of them were vaccinated. The serotype distribution was 19A (67%), 10F and 35A (17%) each. Within the antimicrobial resistance patterns, serotype 19A was multidrug resistant (resistance to: beta-lactams, macrolides, lincosamides and TMP/SMX).

## Conclusion

*S. pneumoniae* continues to be a leading cause of AOM in our country. Serotype 19A accounts for 67% of these infections and exhibits a multidrug-resistant pattern similar to that observed in invasive pneumococcal disease. These findings are consistent with international data and provide a baseline for tracking future AOM trends related to *S. pneumoniae* after the introduction of the PCV-13 vaccine.

---

## Introduction

Acute otitis media (AOM) is a common childhood infection and a leading cause of antibiotic prescriptions [1]. It is estimated that 80% of children experience at least one episode of AOM before the age of three and 50% before the age of one, with a peak occurrence between 9 and 15 months of age [2]. In Latin America, the incidence of AOM ranges from 4.25% to 6.78% [3]. For instance, in a 2016 study published in a Caribbean city in Colombia, the incidence of AOM was found to be 29.4 per 1,000 children [4]. This condition poses significant challenges to public health and health economics, emphasizing the need for effective preventive strategies.

Different pathways are related to AOM development. Although still controversial, bacterial colonization is considered a prerequisite [5–8]. The three most relevant bacterial pathogens reported worldwide are: *Streptococcus pneumoniae*, responsible for 30% to 50% of all cases [9–11], non-typeable *Haemophilus influenzae* (15% to 30%), and *Moraxella catarrhalis* (3% to 20%) [12,13]. In Colombia, similar trends have been observed; the study of Sierra-Lopez *et al* conducted in Cali reported that the most commonly isolated bacteria in pediatric otitis media cases were *H. influenzae* (31%) and *S. pneumoniae* (30%) [14].

Among its complications, otomastoiditis is the most prevalent, although severe conditions such as meningitis and encephalitis have also been documented [15–17]. Middle ear fluid (MEF) analysis plays a critical role in identifying the etiologic agents of AOM. MEF samples provide valuable insights into pathogen prevalence, serotype distribution, and antimicrobial resistance patterns, offering a reliable basis for evaluating the impact of vaccination and guiding therapeutic approaches [18].

Pneumococcal conjugate vaccines (PCV) have been shown to prevent both non-invasive and invasive pneumococcal disease (IPD), including AOM [19–22]. Several studies have reported significant declines in AOM caused by vaccine-included serotypes following the introduction of PCV-7 and PCV-13 in upper-middle-income countries, although non-vaccine serotypes like 19A exhibited replacement effects [20,23,24]. In Colombia, PCV-7 was introduced in 2006 for high-risk children under 2 years of age and its use was expanded in Bogotá in 2008. In 2009, coverage

extended to high-mortality regions, and between 2010 in and 2011, PCV13 replaced PCV7 nationwide. In 2011, PCV10 was incorporated into the national immunization program and became universally available in 2012 reaching 89% coverage [25,26]. The introduction of PCV-10 led to an 84.7% national reduction in IPD due to vaccine serotypes in children under five years of age [27,28]. By 2019, national vaccination coverage reached >90% although significant variability was observed among departments; for instance, in Bolívar, *S. pneumoniae* vaccination historically lagged behind national rates, reaching 80–89% coverage in 2021 [29]. In 2022, PCV-10 was subsequently replaced by PCV-13 [30].

Regarding AOM, a study in Bogotá D.C. and Medellín observed reductions in its incidence following the implementation of PCV-10, consistent with trends reported in other Latin American regions [31]. However, in 2016, an evaluation conducted in Cartagena determined the effectiveness of PCV-10 in a nested case-control study within a cohort of newborns followed up to 18 months of age and found a protection rate of 33% against AOM. The isolated microorganisms were β-lactamase-negative non-typeable *H. influenzae* (20%), *S. pneumoniae* (15%), and serotype 6C as the only pneumococcal recognized serotype. *Streptococcus pyogenes* and *Staphylococcus aureus* were also isolated, corresponding to 10% of the cases, respectively [32].

Although these findings align with global data on the main etiologic agents of bacterial AOM and serotype distribution trends of *S. pneumoniae* [13], critical gaps persist in understanding the regional dynamics in Latin America, particularly in Colombia. It is essential to understand the current role of *S. pneumoniae* in the bacterial etiology of AOM and how the introduction of PCV-10 in infancy impacts the current serotype distribution. The objective of this study was to determine the prevalence of bacterial AOM caused by *S. pneumoniae* in children aged 3–59 months in a Colombian Caribbean city. The study also aimed to identify *S. pneumoniae* serotype distribution and antimicrobial resistance patterns from middle ear fluid (MEF) samples. By identifying prevalent serotypes and their antimicrobial resistance, our data can contribute to optimizing antibiotic stewardship and informing the development of regional public health policies aimed at reducing the burden of AOM in children.

## Materials and methods

This prospective observational cohort single-center study included children aged 3–59 months diagnosed with AOM who consulted the Napoleon Franco Pareja Children's Hospital in Cartagena, Colombia. The recruitment period was from August 5, 2022, to August 5, 2023. Fig 1 summarizes the recruitment process and procedures performed for each case included in the study.

### Ethical considerations

This study was reviewed and approved by the research ethics committee of the Instituto Médico de Alta Tecnología (IMAT) under minute no. 435. All study procedures were implemented in compliance with the Helsinki Declaration. Since participants were underage, written informed consent was obtained from the parent/legal guardian after informing them about the research, in the presence of two witnesses. A copy of the signed and dated consent form was given to the subject before participation in the study (S1 Text). The informed consent adhered to IRB/ERC requirements, applicable laws and regulations. According to Resolution No. 8430 of 1993 of the Colombian Ministry of Health and Social Protection, this study is considered minimum risk [33], since the samples were taken in the context of the medical care of a patient with a clinical diagnosis of AOM and allowed an accurate diagnosis of the etiological agent.

### Population and selection criteria

The study included children of both sexes, aged 3–59 months, diagnosed with AOM, who sought consultation at the Napoleón Franco Pareja Children's Hospital. This hospital is the only pediatric facility in the Colombian Caribbean region that provides comprehensive health services of medium and high complexity to both the contributory (population with ability to pay) and subsidized (population in poverty) regimes of the health system [34].

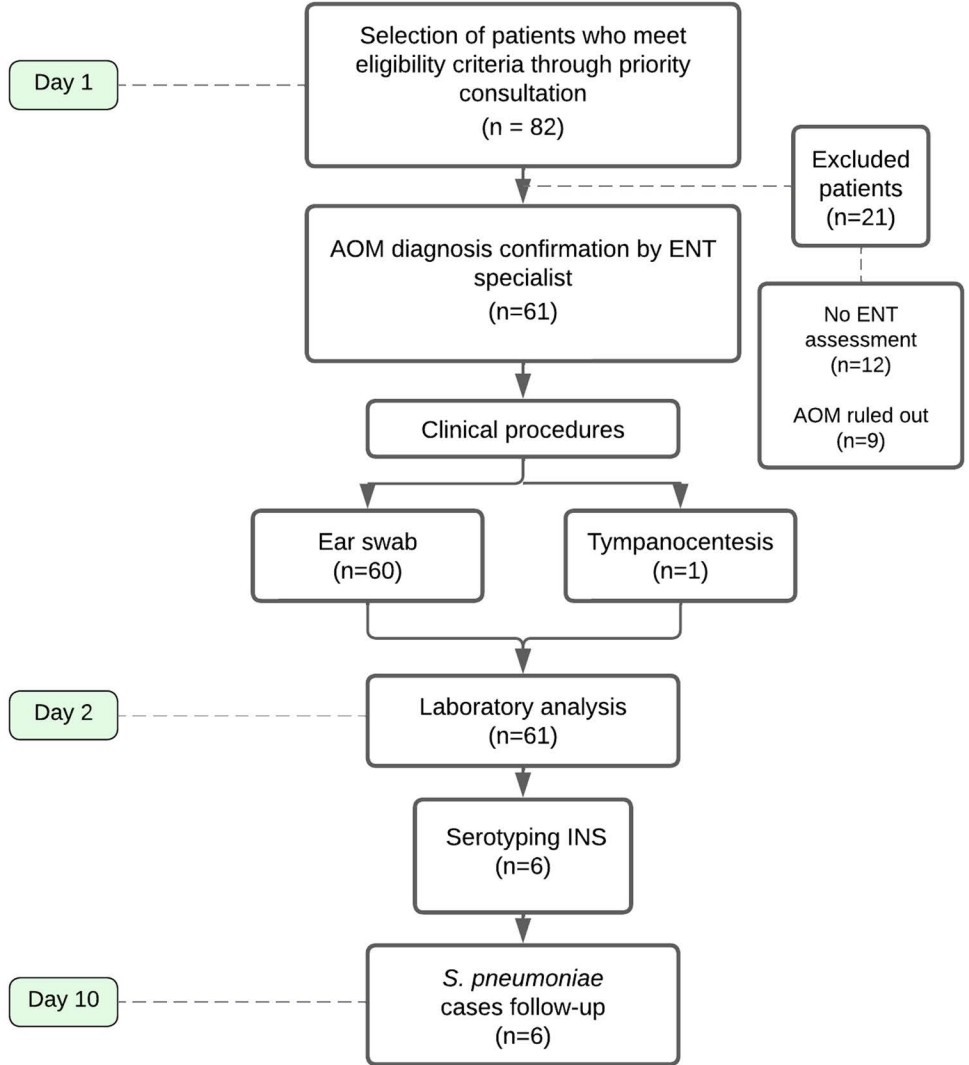

**Fig 1. Overview of the study's workflow.** AOM: acute otitis media; ENT: ear, nose, and throat; INS, Colombian National Health Institute.

A two-step evaluation process was used to define eligibility. Suspected AOM cases were initially screened by general physicians. The screening criteria included: 1) moderate or marked bulging of the tympanic membrane (TM), or mild bulging accompanied by marked erythema and reduced or absent mobility, or the presence of cloudy or purulent otorrhea and rupture of a previously intact TM (within 48 hours); and/or 2) signs and symptoms of acute infection, such as otalgia, decreased hearing, fever, lethargy, irritability, anorexia, vomiting, or diarrhea, with onset within the previous 72 hours. Children meeting these criteria were referred for confirmatory evaluation by an otorhinolaryngologist, who established the final diagnosis based on otoscopic findings and the diagnostic standards of the American Academy of Pediatrics (AAP). The presence of systemic symptoms served as supportive evidence but was not sufficient alone to confirm a case [35].

Only children with otorhinolaryngologist-confirmed AOM were assessed against exclusion criteria and subsequently enrolled in the study. Exclusion criteria for participants in this study included: a) having received systemic antibiotic treatment for a disease other than AOM within 72 hours prior to inclusion; b) having received antimicrobial prophylaxis

for recurrent AOM; and c) having underlying conditions such as progressive neurological disease, acquired or congenital immunodeficiency, cleft palate, chronic TM perforation, or craniofacial anomalies. Additionally, participants with a history of recurrent AOM, defined as a new episode of AOM occurring 3 weeks after initial diagnosis, were also excluded.

### Sample collection

The otorhinolaryngologist evaluated participants who met the inclusion criteria and determined the need for tympanocentesis as a diagnostic and therapeutic procedure in cases with an intact TM using pneumatic otoscopy. In cases of ruptured TM accompanied by spontaneous otorrhea, a swab sample of MEF was taken.

An otorhinolaryngologist performed tympanocentesis under appropriate conditions using a CDT speculum adapted to the patient's age and with prior administration of topical anesthetic. The sample was labeled and stored in a sterile container with the subject's identification number, sample site (left or right ear), and date of collection. It was then sent directly to the clinical laboratory at room temperature.

For MEF sampling, the external auditory canal (EAC) was cleaned with a disposable gauze. Subsequently, a sample was taken using a slit head otoscope with a slit for a thin, flexible wire swab, ensuring no contact with the walls of the EAC [36,37]. In cases of bilateral AOM with discharge, the MEF sample was taken from only one ear. The samples were labeled with the subject's identification number, sample site (left or right ear), and date of collection. They were then sent in Amies medium with activated charcoal directly to the microbiology laboratory at room temperature.

### Laboratory analysis

For the cultures, samples were inoculated onto sterile 5% lamb's blood agar and chocolate agar plates. The plates were then incubated at 37°C in a 5% $CO_2$ atmosphere for 24 hours. After incubation, the colonies were evaluated for growth.

Colonies that had produced alpha-hemolysis were subjected to Gram staining, followed by catalase and oxidase tests.

Phenotypic tests were performed such as: 1) The optochin test: the strains that showed alpha-hemolysis were inoculated on sheep blood agar in a CO2 incubator (5–10%) and a 5 mg optochin disk was placed and incubated for 18–24 hours at a temperature of 37°C. After this, the strains were reviewed and those that had an inhibition halo > 14 mm (sensitive to optochin) confirmed the presence of pneumococcus. 2) The other phenotypic test used was bile solubility: The isolated strains were exposed to bile salts, and these were lysed or destroyed, giving a positive bile solubility test.

The identification of the microorganism and its susceptibility profile were performed with automated methods (Microscan Autoscan-4 System Beckman Coulter®, Pasadena, California, United States). Under this system, the MICroSTREP plus® panel was used to identify the microorganism and its susceptibility to different groups of antimicrobials by means of cards with different minimum inhibitory concentration (MIC): beta-lactams (penicillins, cephalosporins and carbapenems), macrolides (erythromycin, azithromycin and clarithromycin), lincosamides (clindamycin), trimethoprim/sulfamethoxazole (TMP/SMX), chloramphenicol, glycopeptides (vancomycin), and oxazolidinones (linezolid) or sensitive (no resistance to any antimicrobial was reported).

*S. pneumoniae* isolates were prepared in Amies transport medium with activated charcoal and sent to the laboratory of the Colombian National Institute of Health (INS) at a temperature between 18 and 25°C. Serotyping was performed by a phenotypic confirmation method using the capsular swelling test known as the Quellung reaction (Sattens Serum Institute, Copenhagen, Denmark).

### Follow-up

Cases positive for *S. pneumoniae* were followed up with a telephone call after 10 days to evaluate clinical improvement, the need for further consultation due to persistent symptoms, hospitalization, changes in antibiotics, and/or the presence of complications.

## Data analysis

The study examined several sociodemographic variables, including age, ethnic group, parental education level, household structure, and potential risk factors such as exposure to tobacco smoke, daycare attendance, and breastfeeding practices. Regarding clinical variables, we collected information on comorbidities, previous episodes of AOM, and pneumococcal vaccination status. We reviewed the vaccination card and checked PAIWEB, an Expanded Plan of Immunization (EPI) tool to manage and monitor citizens' vaccination history information in the country, which is available at http://www.paiweb.gov.co. We collected all variables of interest using an electronic questionnaire designed on the KoboToolbox® platform.

We estimated the prevalence of AOM caused by *S. pneumoniae* and the distribution of serotypes. Resistance patterns were identified from the MEF samples collected, based on the classification of antimicrobial groups mentioned in the laboratory analysis section. The number and groups of antimicrobials to which each isolated serotype presented resistance were described in relation to the therapeutic management received and the outcome reported in the follow-up.

Descriptive statistical methods were used to summarize sociodemographic and clinical information. For categorical variables, we calculated measures of absolute and relative frequencies. For quantitative variables, we applied a normality test and calculated measures of central tendency and variability, including mean, standard deviation (SD), median, interquartile range (IQR), and percentiles. As an exploratory analysis, the population-level incidence of AOM due to *S. pneumoniae* in the population was estimated using as a population base the projected number of children in Cartagena in 2022, as reported by the National Administrative Department of Statistics (DANE), along with the number of AOM cases reported in Cartagena through the Integrated Social Protection Information System (SISPRO), and an extrapolation of the number of cases attributable to *S. pneumoniae*, based on the prevalence obtained in this hospital-based study. The incidence rate was estimated with 95% confidence intervals. All analyses were conducted using R software version 4.1.2 via the RStudio interface.

## Results

During the study period, a total of 82 patients met eligibility criteria through a priority consultation, but only 61 patients had confirmation of AOM diagnosis by an ENT specialist. The latter group formed the study sample (Fig 1). The peak recruitment months were September (n = 12) and October (n = 8) in 2022, and April 2023 (n = 8) (Fig 2).

The median age of the participants (n = 61) was 12 months (IQR = 8.0–24.0), and 35/61 were male (57.4%). All participants belonged to the subsidized healthcare regime. Of these, 38/61 (62.3%) of parents or caregivers had completed high school. The median number of people per household was 4 (IQR = 4–5), and they lived with 1–3 children. In the study, 6/61 (9.8%) of participants were exposed to tobacco smoke, 13/61 (21.3%) attended day care centers, 32/61 (52.5%) used a bottle, and nearly 50% (31/61) were breastfed for 4–6 months. Only one participant reported more than two episodes of AOM (Table 1).

Regarding breastfeeding practices, 34/55 (61.8%) of infants were exclusively breastfed, with the majority (n = 29/55; 52.7%) being breastfed for 4–6 months. Approximately half of the infants reported the use of a bottle or kettle. In patients with AOM caused by *S. pneumoniae*, the majority were female (n = 4/6; 66.7%) with a median age of 26.5 months (IQR = 8–45). Of these, 4/6 (66.7%) were exclusively breastfed, and none were exposed to tobacco smoke (Table 1).

Regarding the clinical characteristics of the patients (Table 2), of the 61 patients, 38 (62.3%) received the PCV-10 vaccine, as they were born before April 30, 2022. Twenty (32.8%) received the PCV-13 vaccine, and 3 (4.9%) did not receive any pneumococcal vaccine. Among the unvaccinated patients, one tested positive for *S. pneumoniae* and presented with clinical complications. Among the six patients with *S. pneumoniae* isolation, five (83.3%) had received PCV-10, while the remaining case (16.7%) corresponded to the unvaccinated patient mentioned above.

Six of the total participants (9.8%) had at least one previous episode of AOM, with five patients belonging to the group of AOM due to microorganisms other than *S. pneumoniae* and one patient to the group of AOM due to *S. pneumoniae*.

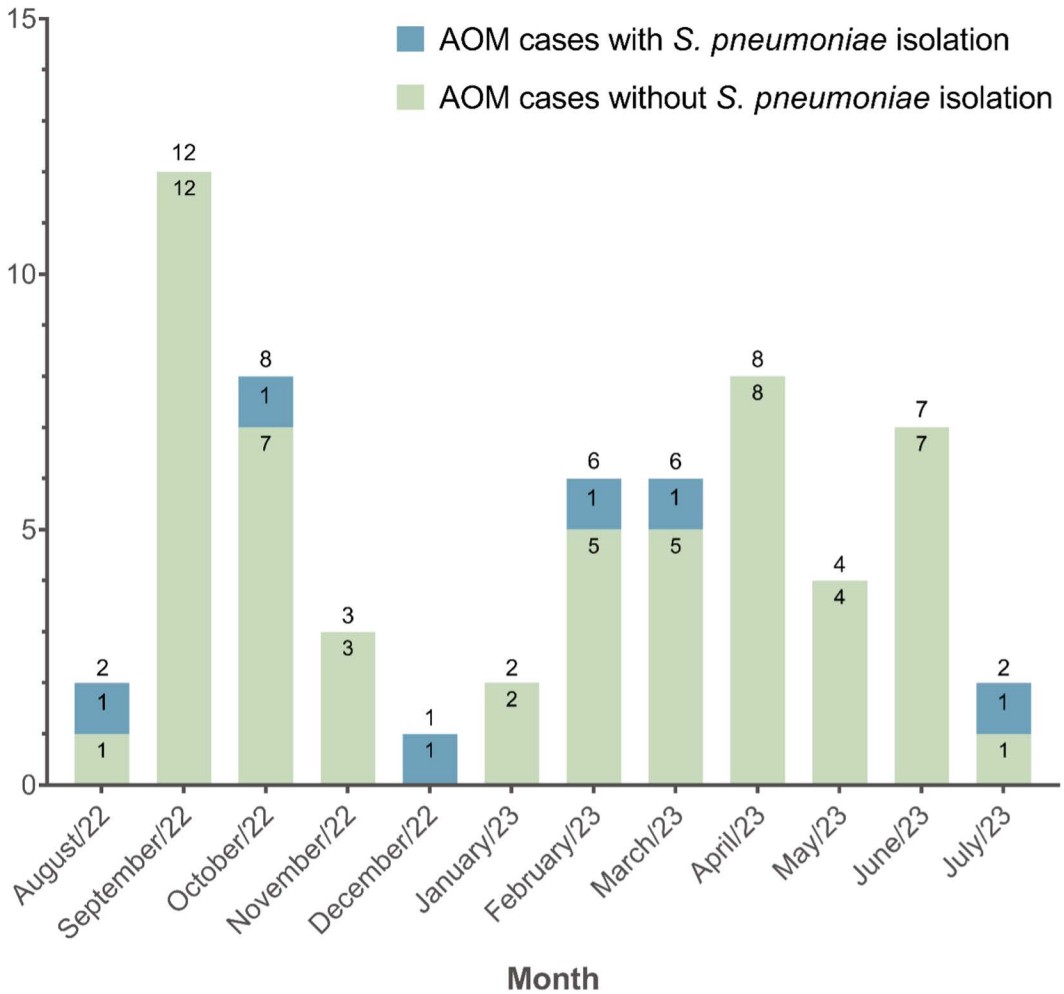

**Fig 2. Inclusion trends of AOM patients.** The monthly number of cases for each group is showed within bars, whereas the total number of patients recruited per month is showed above each bar.

Among the patients, only two had underlying comorbidities. Additionally, 58/61 (95.1%) had received at least one dose of pneumococcal vaccine, and 21/58 (36.2%) had received a booster dose.

Out of the 61 MEF samples collected, *S. pneumoniae* was isolated in six samples, five of which were collected by swabbing and one by tympanocentesis. This results in an estimated prevalence of 9.8%. Based on the information recorded in SISPRO, the incidence rate of AOM in Cartagena is 11.07 (95% CI 10.1–11.8) per 1,000 infants aged 0–4 years. Based on the frequency of *S. pneumoniae* obtained in this study, we estimate 1.09 (95% CI 0.86–1.32) cases per 1,000 person-years in this age group [38]. Fig 3 shows that, in addition to commensal microorganisms of the EAC, the most frequently isolated microorganisms were *Pseudomonas aeruginosa* (n = 9/61; 14.8%), methicillin-resistant *S. aureus* (n = 8/61;13.1%), and *S. pneumoniae* (n = 6/61; 9.8%).

Out of the patients infected with *S. pneumoniae*, five had received the pneumococcal vaccine, while 3/5 (60%) had received a booster dose (Table 2). The patient who had not received the initial dose of the vaccine was a 3-month-old infant (Table 2). Due to the patient's complex clinical outcome, which included established symptoms of AOM concomitant with otomastoiditis, intravenous antibiotics were required, and tympanocentesis was indicated. The serotype isolated from

**Table 1. Sociodemographic characteristics of patients.**

| Variable | All patients | | AOM patients, micro-organism other than *S. pneumoniae* | | AOM patients with *S. pneumoniae* | |
|---|---|---|---|---|---|---|
| | n = 61 | | n = 55 | | n = 6 | |
| | N | % | n | % | N | % |
| **Sex** | | | | | | |
| Female | 26 | 42.6 | 22 | 40.0 | 4 | 66.7 |
| Male | 35 | 57.4 | 33 | 60.0 | 2 | 33.3 |
| **Age (months)** | Median = 12.0 | | Median = 12.0 | | Median = 26.5 | |
| | IQR [8–24] | | IQR [7–24] | | IQR [8–45] | |
| **Education level** | | | | | | |
| Schooled | 12 | 19.7 | 9 | 16.4 | 3 | 50 |
| Unschooled | 49 | 80.3 | 46 | 83.6 | 3 | 50 |
| **Education level of parent/caregiver** | | | | | | |
| None | 1 | 1.6 | 1 | 1.8 | – | – |
| Elementary school, completed | 2 | 3.3 | 2 | 3.6 | – | – |
| Elementary school, not completed | 5 | 8.2 | 5 | 9.1 | – | – |
| High school, completed | 38 | 62.3 | 32 | 58.2 | 6 | 100 |
| High school, not completed | 9 | 14.8 | 9 | 16.4 | – | – |
| Technician or technologist, completed | 2 | 3.3 | 2 | 3.6 | – | – |
| Technician or technologist, not completed | 1 | 1.6 | 1 | 1.8 | – | – |
| University, completed | 2 | 3.3 | 2 | 3.6 | – | – |
| University, not completed | 1 | 1.6 | 1 | 1.8 | – | – |
| **Number of people living in the household** | Median = 4.0 | | Median = 3.0 | | Median = 4.0 | |
| | RIQ [4.0–5.0] | | RIQ [4.0–5.0] | | RIQ [4.0–8.0] | |
| **Number of children living in the household** | Median = 2.0 | | Median = 2.0 | | Median = 2.0 | |
| | RIQ [1.0–3.0] | | RIQ [1.0–3.0] | | RIQ [1.0–3.0] | |
| **Exposure to cigarette smoke** | | | | | | |
| Yes | 6 | 9.8 | 6 | 10.9 | 0 | |
| No | 55 | 90.2 | 49 | 89.1 | 6 | 100 |
| **Attendance to day care** | | | | | | |
| Yes | 13 | 21.3 | 11 | 20.0 | 2 | 33.3 |
| No | 48 | 78.7 | 44 | 80.0 | 4 | 66.7 |
| **Use of baby feeder/bottle** | | | | | | |
| Yes | 29 | 47.5 | 26 | 47.3 | 3 | 50 |
| No | 32 | 52.5 | 29 | 52.7 | 3 | 50 |
| **Exclusive breastfeeding** | | | | | | |
| Yes | 38 | 62.3 | 34 | 61.8 | 4 | 66.7 |
| No | 23 | 37.7 | 21 | 38.2 | 2 | 33.3 |
| **Duration of breastfeeding** | | | | | | |
| < 1 month | 6 | 9.8 | 5 | 9.1 | 1 | 16.7 |
| 1 - 3 months | 9 | 14.8 | 7 | 12.7 | 2 | 33.3 |
| 4 - 6 months | 31 | 50.8 | 29 | 52.7 | 2 | 33.3 |
| 7 - 12 months | 8 | 13.1 | 8 | 14.5 | 0 | 0 |
| > 12 months | 7 | 11.5 | 6 | 10.9 | 1 | 16.7 |

**Table 2. Clinical characteristics of patients.**

| Variable | All patients | | AOM patients, microorganism other than *S. pneumoniae* | | AOM patients with *S. pneumoniae* | |
|---|---|---|---|---|---|---|
| | n = 61 | | n = 55 | | n = 6 | |
| | N | % | n | % | N | % |
| **Previous AOM episodes** | | | | | | |
| 0 | 55 | 90.2 | 50 | 90.9 | 5 | 83.3 |
| 1 | 4 | 6.6 | 3 | 5.5 | 1 | 16.7 |
| 2 | 1 | 1.6 | 1 | 1.8 | – | – |
| > 2 | 1 | 1.6 | 1 | 1.8 | – | – |
| **Comorbidities** | | | | | | |
| Yes | 2 | 3.3 | 2 | 3.6 | – | |
| No | 59 | 96.7 | 53 | 96.4 | 6 | 100 |
| **Received antibiotics 30 days before study inclusion** | | | | | | |
| Yes | 1 | 1.6 | 1 | 1.8 | 0 | – |
| No | 60 | 98.4 | 54 | 98.2 | 6 | 100 |
| **Pneumococcal vaccination** | | | | | | |
| Yes | 58 | 95.1 | 53 | 96.4 | 5 | 83.3 |
| No | 3 | 4.9 | 2 | 3.6 | 1 | 16.7 |
| **Vaccine doses received*** | | | | | | |
| 1 | 4 | 6.9 | 4 | 7.5 | 0 | |
| 2 | 33 | 56.9 | 31 | 58.5 | 2 | 40 |
| Booster dose | 21 | 36.2 | 18 | 3.4 | 3 | 60 |

*Proportion calculated from the total number of vaccinated patients.

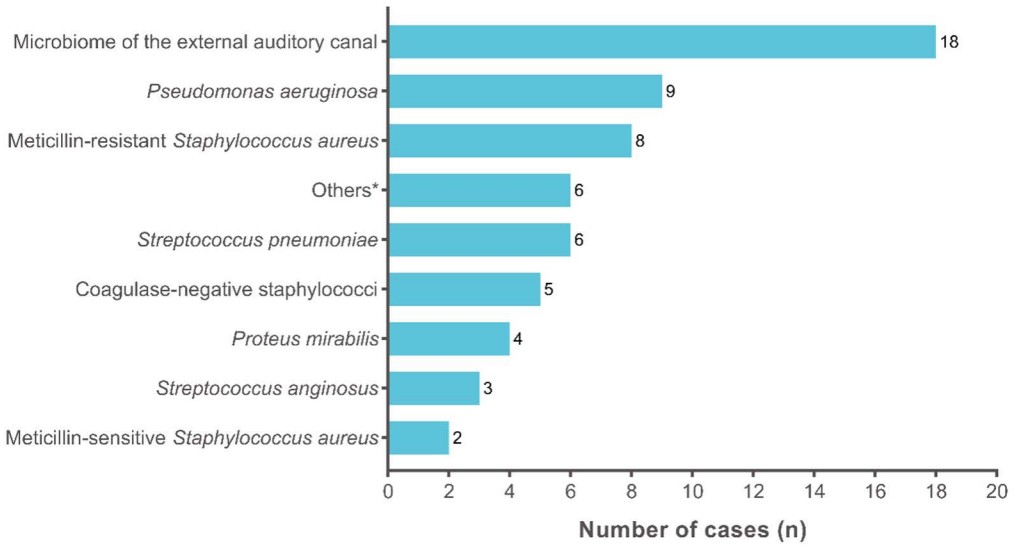

*Other isolated microorganisms: *Streptococcus salivarius*, group A *Streptococcus*, *Micrococcus*, *Enterococcus faecium*, *Escherichia coli* y contaminating non-sporulated Gram-positive bacillus.

**Fig 3. Microorganisms isolated from MEF cultures (n = 61 isolations).**

the MEF in this patient was 19A. In this group of six patients with *S. pneumoniae* isolation, the identified serotypes corresponded to the non-PCV-10 serotype group. Serotype 19A was the most frequent (n = 4), while serotypes 10F (n = 1) and 35A (n = 1) were also isolated.

Regarding resistance patterns, Fig 4 indicates that all pneumococcal serotypes isolated showed some degree of antimicrobial resistance. Serotype 19A presented resistance to up to four groups of antimicrobials, while serotypes 10F and 35A reported resistance to only one group of antimicrobials.

Follow-up of cases diagnosed with AOM due to *S. pneumoniae* (Table 3) showed that one patient had recurrence of symptoms and required a change of antibiotic treatment. The other five cases with isolation of *S. pneumoniae* were effectively treated with first-line antibiotics (i.e., amoxicillin). Intravenous ceftriaxone was used in the patient who required tympanocentesis due to aggressive clinical instauration; however, none of these patients required additional interventions nor had sequelae in the clinical follow-up.

## Discussion

This study provides updated microbiological evidence on AOM in infants from a subsidized healthcare population in Cartagena, Colombia during a period of high PCV-10 coverage. Among 61 children with otorhinolaryngologist-confirmed AOM, *S.*

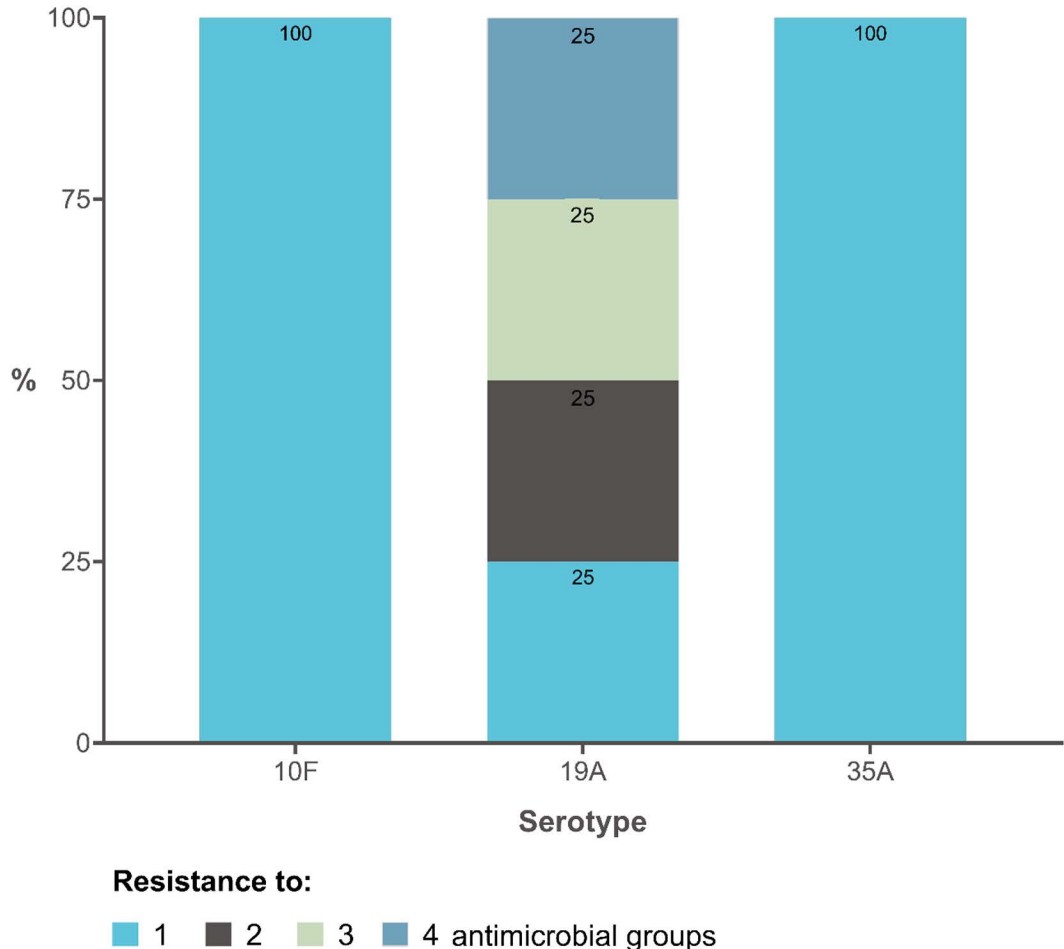

**Fig 4. Resistance pattern of *S. pneumoniae* serotypes to antimicrobial groups (n = 6 isolations).**

**Table 3. Outcome of cases with *S. pneumoniae* isolation.**

| Case # | Isolated serotype | Resistance profile | MIC | Antibiotic treatment | Treatment outcome |
|---|---|---|---|---|---|
| 1 | 10F | Resistance to chloramphenicol | 8 | Oral amoxicillin | Symptoms resolution |
| 2 | 19A | Resistance to cefuroxime, erythromycin, clindamycin, TMP/SMX | 8 >4 >2 >2/38 | Oral amoxicillin | Symptoms resolution |
| 3 | 35A | Resistance to TMP/SMX | >2/38 | Oral amoxicillin | Symptoms resolution |
| 4 | 19A | Resistance to erythromycin | >1 | Oral amoxicillin | Symptoms resolution |
| 5 | 19A | Resistance to macrolide and clindamycin | >=1 | Oral amoxicillin | Symptoms recurrence |
| 6 | 19A | Resistance to cefuroxime, erythromycin, TMP/SMX | >8 >4 >2/38 | IV ceftriaxone | Complicated with otomastoiditis that responded adequately to treatment |

IV, intravenous

*pneumoniae* was identified in 9.8% of MEF samples, with serotype 19A being the most frequent. Notably, all pneumococcal isolates corresponded to non-PCV-10 serotypes, and most showed resistance to at least one antibiotic group. These findings highlight serotype replacement and antimicrobial resistance in the post pneumococcal vaccination era, underscoring the need for enhanced surveillance and potentially broader vaccine formulations tailored to local epidemiology.

The low frequency of *S. pneumoniae* in our study (9.8%) contrasts with the study of Sierra et al conducted in Colombia in 2008, reporting 30% of prevalence, in which only 29% of the children had received the PCV7 vaccine [14]. The reduction aligns with international data showing a decline in pneumococcal AOM, as well as IPD, following the implementation of conjugate vaccines [27]. However, the predominance of serotype 19A (67%) in our findings is consistent with serotype replacement effects observed in Latin America after PCV-10 introduction [39]. This suggests that despite vaccination efforts, non-vaccine serotypes remain an important cause of AOM in older children. Evidence of a similar behavior in Colombia regarding isolates of *S. pneumoniae* causing IPD was found, wherein 19A and 6C are predominant serotypes [26,29]. Notably, the one unvaccinated child developed a severe complication (otomastoiditis), reinforcing the benefits of vaccination despite serotype shifts. As our study, similar finding and complication was reported in Mexico, wherein the majority of cases of otomastoiditis per month (0.15) occurred in unvaccinated participants [15].

Our results are in agreement with the observation that pneumococcal vaccines cause a replacement effect or selective pressure that change the predominance of circulating serotypes, as evidenced in IPD but also in non-invasive pneumococcal disease [40]. The findings of AOM-causing pneumococcal serotypes not included in PCV-10 (19A, 10F, and 35A) align with serotype replacement trends observed in Latin America post-PCV-10 introduction [39]. In contrast, data from Colombia prior to PCV-10 implementation reported serotypes such as 6B, 14, and 19F as predominant causes of AOM [14]. This trend is consistent with findings from other Latin American countries where serotype 19A emerged following PCV-10 implementation. Similarly, Latin American countries with PCV13 schemes recorded a lower frequency of serotype 19A isolation [39]. Serotype 19A accounted for most of the pneumococcal isolates in our study, which is of concern because of its multidrug resistance pattern [41], which can lead to treatment failures and increased complications [42]. Our finding of cases with resistance to macrolides, clindamycin, and TMP/SMX is consistent with regional reports [29]. It is expected that the introduction of PCV13 would also change the distribution of Spn serotypes (serotype replacement) in Colombia [43]. Therefore, continuous epidemiological surveillance during following years is required to evaluate the effectiveness of the current vaccination strategy.

While *S. pneumoniae* was one of the main pathogens, the most frequently isolated pathogens were *P. aeruginosa* (14.8%) and methicillin-resistant *S. aureus* (13.1%). Surprisingly, no isolates of *H. influenzae* were found, despite its high

prevalence and known resistance [44,45]. The high prevalence of commensal or non-typeable isolates (29.5%) may reflect viral infections as etiological agents. MEF cultures are often negative—up to 47% in confirmed cases— [46]. Simultaneous isolation of respiratory viruses may contribute to the pathogenesis of AOM even without bacterial colonization of the nasopharynx. Therefore, the association between symptoms and viral presence in the ear is often significant [47,48].

Regarding seasonality, two peaks of AOM cases were found (Feb–Apr and Sep–Oct), partially overlapping with acute respiratory infections (ARI) trends in the country, where the highest proportion occurred between June and September [49]. This temporal overlap may suggest an epidemiological link, since ARI tends to preceded AOM by predisposing the middle ear to bacterial superinfection [50]. Therefore, the increase in AOM cases observed after ARI peak periods may reflect this pathophysiological sequence, although our data are not sufficient to confirm a consistent seasonal pattern for bacterial AOM.

As vaccine coverage expands to more serotypes, the replacement effect by non-vaccine serotypes [43] or other microorganisms will continue, as will the antimicrobial pressure due to the indiscriminate use of antibiotics. Among the cases of non-pneumococcal AOM, it was found that 25.5% of the participants received the PCV-13 vaccination schedule. This observation could serve as a baseline for future follow-ups to assess the effect of replacement by other microorganisms and serotypes. Pneumococcal surveillance to non-invasive disease, including serotyping and resistance profiling, is critical for tracking the community-level circulation of multidrug-resistant strains. We acknowledge the development and licensure of newer pneumococcal conjugate vaccines (PCV10-SII, PCV-15 and PCV-20 which offer broader serotype coverage and are under evaluation or currently used in various immunization programs of other countries [51–53], but these vaccines were not yet licensed in Colombia at the time of this study.

This study has several strengths. First, the use of high-quality microbiological techniques and standardized procedures ensured the reliability of our findings. The MEF samples were analyzed using validated laboratory methods, including bacterial cultures, optochin sensitivity testing, and bile solubility testing for pneumococcal identification. Additionally, all laboratory procedures were double-checked and cross-verified between the hospital microbiology laboratory and the Colombian INS, ensuring accuracy and consistency. Finally, strict adherence to sample collection and handling protocols minimized contamination risks and ensured high-quality data. The immunization status of cases was verified in a dual manner: first, in priority consultation and then corroborated by the information available in the electronic media of the Colombian EPI. This allowed for the identification and execution of a descriptive analysis of the patients vaccinated with PCV-10 ten years after its implementation in the country in 2012. These strengths ensure the quality of the results, which contribute to the understanding of the impact of vaccination and decision-making.

However, it is important to note that our study has limitations. It was conducted in a single healthcare institution and may not reflect the distribution of pathogens, or the characteristics of subjects treated at home or in other healthcare settings. The evaluation of outcomes through telephone follow-up of only cases of *S. pneumoniae* infection did not provide insight into the clinical course of AOM caused by other etiologic agents. This would have allowed for comparisons of resistance patterns and clinical outcomes among the different microorganisms. Due to the low number of cases of AOM and *S. pneumoniae* infection, there was insufficient statistical power to calculate measures of association between the resistance pattern and recurrence of symptoms and treatment failure. Finally, our results provide a baseline for future larger-scale, multi-center studies and contribute valuable microbiological surveillance data on *S. pneumoniae* serotypes post-PCV-10 introduction.

## Conclusions

This study is relevant to understand the effects of vaccination against *S. pneumoniae* in Colombia, 10 years after the introduction of PCV-10. The prevalence of AOM due to *S. pneumoniae* has been reduced, although the most frequently found serotypes are not included in PCV-10. Notably, serotype 19A caused 67% of the isolates and has an MDR pattern. Although this study is not nationally representative, the results are consistent with those of national and international studies. As Colombia transitions from PCV-10 to PCV-13, this study provides a valuable baseline for monitoring changes in serotype distribution and resistance patterns by *S. pneumoniae* in the context of a non-invasive disease.

## Supporting information

**S1 Text. Informed consent form.**
(DOCX)

## Acknowledgments

The authors express their gratitude to the administrative and assistance personnel of the Napoleón Franco Pareja Children's Hospital 'La Casa del Niño' for their support and contribution to this research. Special thanks to the group of physicians from the priority consultation and emergency department, as well as the residents of the otorhinolaryngology service, for their assistance in recruiting the patients included in the study. Additionally, we would like to acknowledge the methodological advisors of the project.

## Author contributions

**Conceptualization:** Wilfrido Coronell-Rodriguez, Josefina Zakzuk, Nelson R. Alvis-Zakzuk, Nelson Alvis Guzman.

**Data curation:** Rosanna Camerano, Juan Carlos Alvarado-Gonzalez.

**Formal analysis:** Rosanna Camerano, Juan Carlos Alvarado-Gonzalez, Lina Moyano-Tamara.

**Funding acquisition:** Sebastian Medina, Claudia Beltran, Maria Betancur, Monica Rojas, Luis Enrique Farias.

**Methodology:** Wilfrido Coronell-Rodriguez, Alejandra Puerto, Josefina Zakzuk, Nelson R. Alvis-Zakzuk, Sebastian Medina, Claudia Beltran, Maria Betancur, Monica Rojas, Luis Enrique Farias, Nelson Alvis Guzman.

**Project administration:** Nelson R. Alvis-Zakzuk, Lina Moyano-Tamara, Sebastian Medina, Claudia Beltran, Maria Betancur, Monica Rojas, Luis Enrique Farias.

**Resources:** Hernando Pinzon-Redondo.

**Supervision:** Nelson R. Alvis-Zakzuk, Lina Moyano-Tamara, Sebastian Medina, Claudia Beltran, Maria Betancur, Monica Rojas, Luis Enrique Farias.

**Validation:** Wilfrido Coronell-Rodriguez, Juan Carlos Alvarado-Gonzalez, Alejandra Puerto, Hernando Pinzon-Redondo, Perla Villamor, Steven Osorio.

**Visualization:** Rosanna Camerano, Juan Carlos Alvarado-Gonzalez.

**Writing – original draft:** Wilfrido Coronell-Rodriguez, Rosanna Camerano, Juan Carlos Alvarado-Gonzalez, Josefina Zakzuk, Nelson R. Alvis-Zakzuk, Lina Moyano-Tamara, Perla Villamor, Steven Osorio, Nelson Alvis Guzman.

**Writing – review & editing:** Wilfrido Coronell-Rodriguez, Rosanna Camerano, Juan Carlos Alvarado-Gonzalez, Alejandra Puerto, Josefina Zakzuk, Nelson R. Alvis-Zakzuk, Lina Moyano-Tamara, Sebastian Medina, Claudia Beltran, Maria Betancur, Monica Rojas, Luis Enrique Farias, Hernando Pinzon-Redondo, Perla Villamor, Steven Osorio, Nelson Alvis Guzman.

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
