## [Decision Letter · Decision Letter 0]

Dear Dr. Alvis Guzman,

Thank you for submitting your manuscript to PLOS ONE. After careful consideration, we feel that it has merit but does not fully meet PLOS ONE’s publication criteria as it currently stands. Therefore, we invite you to submit a revised version of the manuscript that addresses the points raised during the review process.

We look forward to receiving your revised manuscript.

Kind regards,

Luis Felipe Reyes, M.D., Ph.D., MSc.

Academic Editor

PLOS ONE

Journal Requirements:

3. Thank you for stating the following financial disclosure: “MSD funded this study”.

4. Thank you for stating the following in the Competing Interests section: “I have read the journal's policy and the authors of this manuscript have the following competing interests: Rosanna Camerano, Juan Carlos Alvarado-Gonzales, Alejandra Puerto, Josefina Zakzuk, Nelson R. Alvis-Zakzuk, Lina Moyano-Tamara, Nelson Alvis Guzman received financial support from MSD. Sebastian Medina, Claudia Beltran, Maria Betancur, Monica Rojas, Luis Enrique Farias are employees of Merck Sharp & Dohme Corp., a subsidiary of Merck & Co., Inc., Kenilworth, NJ USA, who may own stock and/or hold stock options in Merck & Co., Inc., Rahway, NJ, US. Other authors have declared that no competing interests exist.”

We note that one or more of the authors are employed by a commercial company: Merck Sharp & Dohme Corp.

a. Please provide an amended Funding Statement declaring this commercial affiliation, as well as a statement regarding the Role of Funders in your study. If the funding organization did not play a role in the study design, data collection and analysis, decision to publish, or preparation of the manuscript and only provided financial support in the form of authors' salaries and/or research materials, please review your statements relating to the author contributions, and ensure you have specifically and accurately indicated the role(s) that these authors had in your study. You can update author roles in the Author Contributions section of the online submission form. Please also include the following statement within your amended Funding Statement. “The funder provided support in the form of salaries for authors [insert relevant initials], but did not have any additional role in the study design, data collection and analysis, decision to publish, or preparation of the manuscript. The specific roles of these authors are articulated in the ‘author contributions’ section.” If your commercial affiliation did play a role in your study, please state and explain this role within your updated Funding Statement.

b. Please also provide an updated Competing Interests Statement declaring this commercial affiliation along with any other relevant declarations relating to employment, consultancy, patents, products in development, or marketed products, etc. Within your Competing Interests Statement, please confirm that this commercial affiliation does not alter your adherence to all PLOS ONE policies on sharing data and materials by including the following statement: "This does not alter our adherence to PLOS ONE policies on sharing data and materials.” (as detailed online in our guide for authors http://journals.plos.org/plosone/s/competing-interests) . If this adherence statement is not accurate and there are restrictions on sharing of data and/or materials, please state these. Please note that we cannot proceed with consideration of your article until this information has been declared. Please include both an updated Funding Statement and Competing Interests Statement in your cover letter. We will change the online submission form on your behalf.

5. In the online submission form, you indicated that “Data cannot be shared publicly because of sensitive information. Data will be shared upon request.”

All PLOS journals now require all data underlying the findings described in their manuscript to be freely available to other researchers, either 1. In a public repository, 2. Within the manuscript itself, or 3. Uploaded as supplementary information. This policy applies to all data except where public deposition would breach compliance with the protocol approved by your research ethics board. If your data cannot be made publicly available for ethical or legal reasons (e.g., public availability would compromise patient privacy), please explain your reasons on resubmission and your exemption request will be escalated for approval.

6. PLOS requires an ORCID iD for the corresponding author in Editorial Manager on papers submitted after December 6th, 2016. Please ensure that you have an ORCID iD and that it is validated in Editorial Manager. To do this, go to ‘Update my Information’ (in the upper left-hand corner of the main menu), and click on the Fetch/Validate link next to the ORCID field. This will take you to the ORCID site and allow you to create a new iD or authenticate a pre-existing iD in Editorial Manager.

7. Please ensure that you refer to Figures 2-4 in your text as, if accepted, production will need this reference to link the reader to the figure.

8. We note you have included a table to which you do not refer in the text of your manuscript. Please ensure that you refer to Table 2 and 3 in your text; if accepted, production will need this reference to link the reader to the Table.

Reviewers' comments:

Reviewer's Responses to Questions

**Comments to the Author**

1. Is the manuscript technically sound, and do the data support the conclusions?

Reviewer #1: Partly

Reviewer #2: Yes

2. Has the statistical analysis been performed appropriately and rigorously?

Reviewer #1: Yes

Reviewer #2: Yes

3. Have the authors made all data underlying the findings in their manuscript fully available?

Reviewer #1: Yes

Reviewer #2: Yes

4. Is the manuscript presented in an intelligible fashion and written in standard English?

Reviewer #1: Yes

Reviewer #2: Yes

Reviewer #1: This is an interesting paper entitled: Evaluation of Streptococcus pneumoniae as a cause of acute otitis media in Colombia: a prospective study. I want to thank the authors for their efforts in researching pneumococci. Even when the paper is well-written and has some important information, in my scope, it looks like a mix of information in different sections. The manuscript has some major aspects that must be addressed before your work is ready for publication. Grammatically it is well-written and understandable. However, it lacks an organized structure; some sections are very long and lose a common thread. A major restructure, and a new version of the discussion is needed.

Structure:

Please number the lines of the document to make it easier to track sentences that need changes or improvements.

Authors Contribution: use the author’s initials for each section

Introduction:

The introduction looks very long and should be reorganized, with a maximum structure of 3 or 4 paragraphs. In some parts, authors seem to be throwing out ideas for discussion about the subject matter, but they are not argued and remain as free sentences. The number of paragraphs of less than 10 lines hinders the reader from jumping to ideas. Provide more information about MEF and its used in previous studies

The initial sentence in paragraph 4 should be modified. Also, reference number 4 does not support the sentence; there are many different pathways related to the development of SPN AOM. SPN colonization is considered a prerequisite, but there is some controversial data in this regard.

Paragraph 5 does not include meningitis as a complication; relevant data about SPN AOM igniting invasive pneumococcal disease is available. this paragraph is weak; it lacks data on vaccination, vaccination coverage, and circulating serotypes. Colombian literature on these topics has been written by authors such as Severiche et al. and Serrano et al.

Paragraph 6 mentioned several studies, but the references do not match. You should revise the metanalysis in this regard. There is so much information in children rather than in adults. The section that refers to the SPN capsular switch needs more data and support.

Paragraph 7, please include the rates of effective vaccination in Colombia with the previous two vaccines. To provide background information to the reader about which vaccine provided “protection” while the study was being performed.

Paragraph 8: Please provide the numbers for the decrease in incidence.

Methods

Please provide clear information about the type of study. Is this a monocentric study? (provide a line saying so )

What kind of guidelines were followed to perform a study on children? Please provide some lines in the first paragraph of methods standing the Helsinki declaration, the informed consent process, and the approval of an ethical committee. Consider moving the ethical considerations section after the first paragraph as this is a study performed in children.

As tympanocentesis is an invasive procedure with several risks, I would like the authors to provide supplemental information a copy of the informed consent.

It is important to include in figure 1 the number of patients screened

How did the authors avoid reporting a false SPN such as Streptococcus pseudonemoniae, which usually shows alpha hemolysis halo and sensitivity to optokine?

how was the value of patients to be included estimated?

Results

There is a typo in the first paragraph of results (after ... April 2023…)

Only 6 SPN cases among 61 patients were recruited. What was the etiology for the other 55 ?

There are several typos repeatedly in the results “(Error! Reference source not found.)”

I suggest that the authors review the reporting structure of the results, use simple numerators and denominators when referring to percentages to make it easier to read, e.g. : XX/XX (XX%)

Discussion

It is ambitious to say that this single-center study of 62 patients describes the effect of vaccination in children suffering from otitis when 1 in 4 children who consult for this possible diagnosis. Likewise, it is estimated that 40% of the consultations are for AOM in children, and this cut-off seems small and even more so when it is mentioned that the hospital where it was carried out is the one that provides the most care to patients from both the private and public sectors.

The discussion must be reorganized and reformulated. The aspects to be discussed should be contrasted with external or internal data previously found before the PCV-10 vaccine introduction.

The strengths of the study should be focused on supporting the results by using high-quality techniques, standard procedures, and double-checked test were performed, not in the team training.

Best wishes , look forward to see a new version of your work !

Reviewer #2: I find this article very interesting, as it contributes to the understanding of the epidemiology of pneumococcus as a cause of acute otitis media in Colombia.

In the introduction, the sentence: “In Latin America, the incidence of AOM ranges from 4.25% to 6.78% [2]. In a 2016 study published in a Caribbean city in Colombia, the incidence of AOM was found to be 29.4% per 1,000 children [3]. This poses a challenge for public health and health economics”

It is recommended that the values be presented in the same units to facilitate comparison, either as percentages or as rates per 1,000 children. Review the text "29.4% per 1,000 children," as the "%" sign is unnecessary.

In the introduction, review the sentence: “during the early introduction of PCV-7/PCV-13 (2003-2005), and in the late period after the introduction of PCV-7/PCV-13 (2006-2009)”.

PCV13 was not available during those time periods.

In the introduction, review the sentence: “In Colombia, the PCV-7 vaccine (Prevnar®) was introduced in the Expanded Program on Immunization (EPI) in Bogotá D.C. in 2009 with a '2+1' schedule at 2, 4, and 12 months of age. It was implemented throughout the country during 2010 and 2011”.

Other studies report different dates for the implementation of PCV7 in Colombia: “In Colombia, vaccination against pneumococcus began in 2006 with PCV7, initially targeting children under 2 years of age at high risk of IPD. Since 2008, the vaccine was administered universally initially in Bogotá, and since 2009 it was extended to the departments of Colombia with the highest mortality from acute respiratory infection. In the 2010–2011 period, given the withdrawal of PCV7 from the market, PCV13 was administered in Bogotá and in the higher risk departments”. Ref: Camacho-Moreno G, Leal AL, Patiño-Niño J et al. Serotype distribution, clinical characteristics, andantimicrobial resistance of pediatric invasive pneumococcal disease in Colombia during

PCV10 mass vaccination (2017–2022). Front. Med 2024, 11:1380125.doi: 10.3389/fmed.2024.1380125.

The implementation dates of PCV10 and PCV13 are the same as those reported in other studies from Colombia.

In results: A total of 61 patients were included in the study. How many cases of otitis media occurred in the hospital during the study period? What proportion of patients who presented with AOM were included in the study?

Figure 1 indicates that 82 patients were candidates; I recommend including this information in the main text.

In results delete the phrases: “Error! Reference source not found.Error! Reference source not found”

I recommend describing in the results the type of pneumococcal vaccine received by the 5 patients who were vaccinated. The phrase: “Of the six cases with S. pneumoniae isolation, five had received PCV-10 given that they were born before April 30, 2022” in the discussion should be moved to the results section.

It is also recommended to describe, in the results section, the vaccines used in patients who experienced non-pneumococcal otitis.

How many patients had negative cultures? In table 2 "55 AOM patients, microorganism other than S. pneumoniae," suggesting that all cultures were positive and in Figure 3, the culture results are described, and it indicates that all cases had microbiological isolation, 18 had microbiome of the external auditory canal. However, the discussion states, "In 37% of the cultures, there was no bacterial growth, a finding similar to our study." Review this phrase.

In results the phrase: “Regarding resistance patterns, Figure indicates that all pneumococcal serotypes isolated showed some degree of antimicrobial resistance” Add the corresponding figure number.

Table 3. Clarify whether the beta-lactam resistance is to penicillin and/or ceftriaxone. If resistant to penicillin or ceftriaxone, it is advisable to report the MIC.

In the discussion section the phrase: “who received PCV-10 previously In addition” Add a period after previously.

In the discussion section the phrase: “The findings of AOM-causing pneumococcal serotypes not included in PCV-10 (19, 10F and 35)” The serotype is 19ª

In the discussion section the phrase: “Our study found that the samples exhibited a resistance pattern consistent with what has been described in the literature. This included resistance to macrolides, clindamycin, TMP/SMX, and aminoglycosides” Pneumococcus is intrinsically resistant to aminoglycosides.

**Do you want your identity to be public for this peer review?** For information about this choice, including consent withdrawal, please see our Privacy Policy

Reviewer #1: No

Reviewer #2: **Yes: ** German Camacho-Moreno

---

## [Author Response · Author response to Decision Letter 1]

13 Mar 2025

Luis Felipe Reyes, MD, MSc. PhD

Academic editor

PLOS ONE

Dear Dr. Reyes and reviewers,

Thank you for your comments and valuable suggestions. We are sure that those have allowed us to improve our manuscript considerably. The modifications introduced in the manuscript were done as required. All recommendations were implemented and are highlighted in red throughout the text. Detailed responses to the journal requirements and reviewer's comments are presented as follows:

R/ Thank you. We have reviewed PLOS ONE's style requirements and have adjusted the manuscript to conform to them.

R/ Funding information was removed from the manuscript

R/ This study was funded by MSD Colombia, a subsidiary of Merck & Co., Inc., Rahway, NJ, USA. Claudia Beltran, Sebastian Medina, Monica Maria Rojas, Maria Alejandra Betancur and Luis Enrique Farias are MSD employees. This did not interfere with the authors' ability to analyze, interpret the data, or prepare the manuscript.

We updated the cover letter considering this new statement.

4. We note that one or more of the authors are employed by a commercial company: Merck Sharp & Dohme Corp.

A. Please provide an amended Funding Statement declaring this commercial affiliation, as well as a statement regarding the Role of Funders in your study. If the funding organization did not play a role in the study design, data collection and analysis, decision to publish, or preparation of the manuscript and only provided financial support in the form of authors' salaries and/or research materials, please review your statements relating to the author contributions, and ensure you have specifically and accurately indicated the role(s) that these authors had in your study. You can update author roles in the Author Contributions section of the online submission form. Please also include the following statement within your amended Funding Statement. “The funder provided support in the form of salaries for authors [insert relevant initials], but did not have any additional role in the study design, data collection and analysis, decision to publish, or preparation of the manuscript. The specific roles of these authors are articulated in the ‘author contributions’ section.” If your commercial affiliation did play a role in your study, please state and explain this role within your updated Funding Statement.

B. Please also provide an updated Competing Interests Statement declaring this commercial affiliation along with any other relevant declarations relating to employment, consultancy, patents, products in development, or marketed products, etc. Within your Competing Interests Statement, please confirm that this commercial affiliation does not alter your adherence to all PLOS ONE policies on sharing data and materials by including the following statement: "This does not alter our adherence to PLOS ONE policies on sharing data and materials.” (as detailed online in our guide for authors http://journals.plos.org/plosone/s/competing-interests). If this adherence statement is not accurate and there are restrictions on sharing of data and/or materials, please state these. Please note that we cannot proceed with consideration of your article until this information has been declared. Please include both an updated Funding Statement and Competing Interests Statement in your cover letter. We will change the online submission form on your behalf.

A. Thank you for this commentary. We already provided and amended the Funding Statement, as follows: “This study was funded by MSD Colombia, a subsidiary of Merck & Co., Inc., Rahway, NJ, USA. Claudia Beltran, Sebastian Medina, Monica Maria Rojas, Maria Alejandra Betancur and Luis Enrique Farias are MSD employees. This did not interfere with the authors' ability to analyze, interpret the data, or prepare the manuscript.”

B. Thanks for your suggestion. We provided an updated Competing Interests Statement, as follows: “This study was funded by MSD Colombia, a subsidiary of Merck & Co., Inc., Rahway, NJ, USA. Claudia Beltran, Sebastian Medina, Monica Maria Rojas, Maria Alejandra Betancur and Luis Enrique Farias are MSD employees. This does not alter our adherence to PLOS ONE policies on sharing data and materials”

5. In the online submission form, you indicated that “Data cannot be shared publicly because of sensitive information. Data will be shared upon request.” All PLOS journals now require all data underlying the findings described in their manuscript to be freely available to other researchers, either 1. In a public repository, 2. Within the manuscript itself, or 3. Uploaded as supplementary information. This policy applies to all data except where public deposition would breach compliance with the protocol approved by your research ethics board. If your data cannot be made publicly available for ethical or legal reasons (e.g., public availability would compromise patient privacy), please explain your reasons on resubmission and your exemption request will be escalated for approval.

R/ We appreciate your commentary in terms of the PLOS data policy. However, sharing the information in a public repository could jeopardize informed consent approved by the Ethics Committee of IMAT Oncomédica (ONC-CEI-CEI-231-2022). Explicitly the informed consent says: “I have also been informed that the tests and procedures will be performed by expert personnel, will be at no cost to me, and their results will be confidential”, which implies that if we potentially share the database, we will be braking the agreement stablished with the patients and their families.

The above support the journal policy which stablish that “This policy applies to all data except where public deposition would breach compliance with the protocol approved by your research ethics board”.

6. PLOS requires an ORCID iD for the corresponding author in Editorial Manager on papers submitted after December 6th, 2016. Please ensure that you have an ORCID iD and that it is validated in Editorial Manager. To do this, go to ‘Update my Information’ (in the upper left-hand corner of the main menu), and click on the Fetch/Validate link next to the ORCID field. This will take you to the ORCID site and allow you to create a new iD or authenticate a pre-existing iD in Editorial Manager.

R/ Thank you for this suggestion. We included the ORCID ID of the corresponding author in the online form. https://orcid.org/0000-0001-9458-864X ORCID NELSON ALVIS GUZMAN

7. Please ensure that you refer to Figures 2-4 in your text as, if accepted, production will need this reference to link the reader to the figure.

R/ Thank you, all figures are referenced in the manuscript.

8. We note you have included a table to which you do not refer in the text of your manuscript. Please ensure that you refer to Table 2 and 3 in your text; if accepted, production will need this reference to link the reader to the Table.

R/ Thank you, all figures and tables were double checked in order to ensure correctly reference into the manuscript.

Reviewers' comments:

Reviewer #1

Manuscript format and Material

1. This is an interesting paper entitled: Evaluation of Streptococcus pneumoniae as a cause of acute otitis media in Colombia: a prospective study. I want to thank the authors for their efforts in researching pneumococci. Even when the paper is well-written and has some important information, in my scope, it looks like a mix of information in different sections. The manuscript has some major aspects that must be addressed before your work is ready for publication. Grammatically it is well-written and understandable. However, it lacks an organized structure; some sections are very long and lose a common thread. A major restructure, and a new version of the discussion is needed.

R/ Thank you for your valuable feedback on our manuscript. We acknowledge your comments regarding the manuscript's structure and the organization of information across different sections. To address this, we carefully revised the manuscript to ensure a more coherent and logical flow. We also worked on restructuring the discussion section to enhance clarity and improve its alignment with the study’s findings.

Introduction

2. The introduction looks very long and should be reorganized, with a maximum structure of 3 or 4 paragraphs. In some parts, authors seem to be throwing out ideas for discussion about the subject matter, but they are not argued and remain as free sentences. The number of paragraphs of less than 10 lines hinders the reader from jumping to ideas. Provide more information about MEF and its used in previous studies

R/ Thank you for your valuable feedback. In response to your comments, we have shortened and reorganized the introduction to improve clarity and focus. The revised introduction is now structured into fewer paragraphs, ensuring a smoother flow and better alignment with the manuscript's objectives. Additionally, we have clarified the role of middle ear fluid (MEF) analysis in AOM research and provided relevant information to address the reviewer's request (Pages 3-4; lines 52-103).

3. The initial sentence in paragraph 4 should be modified. Also, reference number 4 does not support the sentence; there are many different pathways related to the development of SPN AOM. SPN colonization is considered a prerequisite, but there is some controversial data in this regard.

R/ Thank you for recommendation. We have carefully reviewed the references and incorporated new references that better support the statement regarding the pathways involved in SPN AOM development and the role of SPN colonization as a prerequisite, acknowledging the existing controversy in the literature (Pages 3-4; lines 52-103).

4. Paragraph 5 does not include meningitis as a complication; relevant data about SPN AOM igniting invasive pneumococcal disease is available. this paragraph is weak; it lacks data on vaccination, vaccination coverage, and circulating serotypes. Colombian literature on these topics has been written by authors such as Severiche et al. and Serrano et al.

R/ After updating introduction and shorten its length, paragraph 2 mentions meningitis as a complication "Among its complications, otomastoiditis is the most prevalent, although severe conditions such as meningitis and encephalitis have also been documented (9). In paragraph 3, data on vaccination, vaccination coverage, and circulating serotypes is presented (Page 4; lines 70-83).

5. Paragraph 6 mentioned several studies, but the references do not match. You should revise the metanalysis in this regard. There is so much information in children rather than in adults. The section that refers to the SPN capsular switch needs more data and support.

R/ Thank you for your insight. Since introduction has been shortened, the order of paragraphs has changed. Meningitis as a complication is now mentioned in paragraph 2. Paragraph 3 contains relevant and precise information about SPN vaccination coverage. The suggested references have been added to the paragraphs (Page 4; lines 70-83).

6. Paragraph 7, please include the rates of effective vaccination in Colombia with the previous two vaccines. To provide background information to the reader about which vaccine provided “protection” while the study was being performed.

R/ The paragraph has been revised, and we have incorporated vaccination coverage rates to provide a clearer epidemiological context. The revised text now states:

"In 2011, PCV10 was incorporated into the national immunization program and became universally available in 2012, reaching 89% coverage (23,24). The introduction of PCV-10 led to an 84.7% national reduction in IPD due to vaccine serotypes in children under five years of age (25,26). By 2019, national vaccination coverage reached >90%, although significant variability was observed among departments; for instance, in Bolívar, S. pneumoniae vaccination historically lagged behind national rates, reaching 80–89% coverage in 2021." (Page 4; lines 77-83).

7. Paragraph 8: Please provide the numbers for the decrease in incidence.

R/ The paragraph has been revised to ensure accuracy regarding decrease in incidence of IPD in Colombia as follows:

“The introduction of PCV-10 led to an 84.7% national reduction in invasive pneumococcal disease (IPD) due to vaccine serotypes in children under five years of age" (Page 4; lines 78-80).

Methods

8. Please provide clear information about the type of study. Is this a monocentric study? (provide a line saying so )

R/ We appreciated your suggestion. The sentence has been modified, as follows:

“This prospective observational cohort single-center study included children aged 3 to 59 months diagnosed with AOM who consulted the Napoleon Franco Pareja Children's Hospital in Cartagena, Colombia” (Page 5; lines 105-106).

9. What kind of guidelines were followed to perform a study on children? Please provide some lines in the first paragraph of methods standing the Helsinki declaration, the informed consent process, and the approval of an ethical committee. Consider moving the ethical considerations section after the first paragraph as this is a study performed in children.

R/ Thanks, we have moved the ethical considerations section after the first paragraph of methods. Also, we have added this sentence

"All study procedures were implemented in compliance with the Helsinki Declaration" and additional supplementary information about consent informed form (Page 6; lines 110-121).

10. Tympanocentesis is an invasive procedure with several risks, I would like the authors to provide supplemental information with a copy of the informed consent.

R/ We understand your inquiry. However, Tympanocentesis was not routinely done in all cases, it was only performed in one patient who presented clinical complications and who also had positive Streptococcus pneumoniae. Also, we added informed consent form as supporting information (See S1 Text. Informed consent form ) (Page 6; line 116; Page 15 lines 270-272).

11. How did the authors avoid reporting a false SPN such as Streptococcus pseudonemoniae, which usually shows alpha hemolysis halo and sensitivity to optokine?

R/ Thank you very much for this excellent question. Although Streptococcus pseudopneumoniae, is a different species from Streptococcus pneumoniae, shows a similarity of almost 50% with pneumococcus. To avoid reporting false pneumococcus, the following was done:

Phenotypic tests were performed such as:

1. The optochin test: the strains that showed alpha-hemolysis were inoculated on sheep blood agar in a CO2 incubator (5-10%) and a 5 mg optochin disk was placed and incubated for 18-24 hours at a temperature of 37ºC. After this, the strains were reviewed and those that had an inhibition halo > 14 mm (sensitive to optochin) confirmed the presence of pneumococcus, since streptococcus pseudopneumoniae is resistant to optochin when its strains are exposed to a CO2 atmosphere of 5-10%. For this reason, we did not leave the strains incubating with optochin in a normal atmosphere because then streptococcus pseudopneumoniae is sensitive to optochin.

2. The

---

## [Decision Letter · Decision Letter 1]

Dear Dr. Alvis Guzman,

Thank you for submitting your manuscript to PLOS ONE. After careful consideration, we feel that it has merit but does not fully meet PLOS ONE’s publication criteria as it currently stands. Therefore, we invite you to submit a revised version of the manuscript that addresses the points raised during the review process.

We look forward to receiving your revised manuscript.

Kind regards,

Luis Felipe Reyes, M.D., Ph.D., MSc.

Academic Editor

PLOS ONE

Reviewers' comments:

Reviewer's Responses to Questions

**Comments to the Author**

Reviewer #1: All comments have been addressed

Reviewer #2: All comments have been addressed

2. Is the manuscript technically sound, and do the data support the conclusions?

Reviewer #1: Yes

Reviewer #2: Yes

3. Has the statistical analysis been performed appropriately and rigorously?

Reviewer #1: N/A

Reviewer #2: Yes

4. Have the authors made all data underlying the findings in their manuscript fully available?

Reviewer #1: Yes

Reviewer #2: Yes

5. Is the manuscript presented in an intelligible fashion and written in standard English?

Reviewer #1: Yes

Reviewer #2: Yes

Reviewer #1: I want to thank the authors for their effort and dedication. They have responded to the suggestions made in the previous revision, and their manuscript is now a better version.

Despite the improvement of this manuscript, some suggestions persist, and I quote below:

1. In line 52, a reference is missing to support this sentence.

2. The study by Sierra Lopez and Zapata should also be included in the introduction to give specific context to the entity in question.

3. It is important to review the lines referring to the introduction of the vaccines; the flow of information varies from global to local contexts repeatedly and can confuse the reader. I suggest introducing the complete global context and ending with the local context. lines 76-77

4. Line 83 has no reference

5. I understand that for safe handling of sensitive data, the hospital information cannot be accessed; however, in the methodology, they mention that case definitions were made, and then, based on this, inclusion and exclusion criteria were applied. I have two doubts about this: 1. The definition of case 2 includes unspecific symptoms of systemic infection that, in the age of the selected population, encompasses a broad spectrum of entities. It is also unclear whether the definition should include all symptoms or just one. Thus, is why it would be important to show how many cases were defined as case definition 1 o r case definition 2 and then which ones met the inclusion criteria. to modify the figure 1. In case you do not have or can not show the suggested data, explain better the screening and recruitment method. Were all patients who met a case definition (1 or 2) assessed to evaluate inclusion and exclusion criteria, and then it was defined that they entered the study?

6. Regarding MEF sample collection, no references are cited to support the procedure used in this study. Is the sampling protocol proprietary? Was it adapted? I request the authors to provide further information in this regard.

7. Since the authors' results suggest existing differences in the groups' characteristics, it would be worthwhile to analyze the difference through the appropriate statistical tests and add the p-value of this result to Table 1. This would evidence the differences.

8. There appears to be a difference in median age between patients with PNS and those with other causes, which should be discussed further. Could this be related to the increase in transient SPN colonization of the nasopharynx in school children?

9. In the discussion, when referring to the vaccine recommendation, other types of vaccines, such as PCV10-SII (PNEUMOSIL) that protect against SPN serotypes 1, 5, 6A, 6B, 7F, 9V, 14, 19A, 19F, and 23F are not discussed. PCV-15 and PCV-20 are not mentioned either. Likewise, this recommendation does not make sense because the phenomenon of capsular replacement in other countries has already shown that once the vaccine is changed, the circulating serotypes begin to change in the following 5-10 years. Colombia started vaccination with PCV-13 3 years ago after many countries included PCV-15 in their programs. The discussion about the recommendation for the next vaccine should revolve around vaccines after PCV-13. Suggested references PMID (38336559, 39591182, 39153492)

10. Although one of their important findings is that serotype 19a was found in a higher proportion, they do not discuss how, in Colombia, this circulating serotype is also the major cause of IPD PMID (34641809,32964223).

11. There are paragraphs in the discussion that merely mention facts but do not contradict or support the researchers' findings and do not contribute to the scope referred to in the objective. Examples of these are found in lines 344-347, 357-367, and 386-389.

12. I kindly suggest refining the information in the discussion to make the reader understand in a more adequate way the importance of the findings and how this study is the basis for further research on pneumococcal diseases different from IPD, as well as how the impact of the vaccination that has been applied to the Colombian context is reflected in its results and could be modified with the introduction of other vaccines.

I consider the findings of this group of researchers very valuable. I believe this study could be the cornerstone for implementing a program of active surveillance and tracking of causes of acute otitis media. I hope my suggestions can be incorporated into an improved version of their wonderful work.

Reviewer #2: (No Response)

**Do you want your identity to be public for this peer review?** For information about this choice, including consent withdrawal, please see our Privacy Policy

Reviewer #1: No

Reviewer #2: **Yes: ** German Camacho Moreno

---

## [Author Response · Author response to Decision Letter 2]

23 May 2025

Luis Felipe Reyes, MD, MSc. PhD

Academic editor

PLOS ONE

Dear Dr. Reyes and reviewers,

We thank the reviewers for their thorough evaluation and constructive comments. In response, we have carefully revised the manuscript to address all points raised. Below, we detail each reviewer’s comment followed by our response and a summary of the changes made in the revised version of the manuscript.

Comment 1

Reviewer comment:

In line 52, a reference is missing to support this sentence.

Author response:

A reference has been added to the aforementioned line: “Gavrilovici C, Spoială EL, Miron IC, Stârcea IM, Haliţchi COI, Zetu IN, et al. Acute Otitis Media in Children—Challenges of Antibiotic Resistance in the Post-Vaccination Era. Microorganisms. 2022;10(8).”

Comment 2

Reviewer comment:

The study by Sierra Lopez and Zapata should also be included in the introduction to give specific context to the entity in question.

Author response:

We included the study by Sierra et al. in the introduction to provide specific local context regarding the etiology of otitis media in Colombia. The following sentence was added (Lines 64–66): “In Colombia, similar trends have been observed; a study conducted in Cali reported that the most commonly isolated bacteria in pediatric otitis media cases were H. influenzae (31%) and S. pneumoniae (30%) (14).”

Comment 3

Reviewer comment:

It is important to review the lines referring to the introduction of the vaccines; the flow of information varies from global to local contexts repeatedly and can confuse the reader. I suggest introducing the complete global context and ending with the local context. lines 76-77

Author response:

We have reorganized lines 73–87 to improve the clarity and flow of information about the introduction of pneumococcal vaccines. The section now begins with the global perspective and then moves on to the national context in Colombia.

Comment 4

Reviewer comment:

Line 83 has no reference.

Author response:

Reference 29 was added: Rodríguez WC, Mora-Salamanca AF, et al. Pediatric invasive pneumococcal disease in Bolívar, Colombia. Infez Med. 2024;32(4):506–17.

Comment 5

Reviewer comment:

I understand that for safe handling of sensitive data, the hospital information cannot be accessed; however, in the methodology, they mention that case definitions were made, and then, based on this, inclusion and exclusion criteria were applied. I have two doubts about this: 1. The definition of case 2 includes unspecific symptoms of systemic infection that, in the age of the selected population, encompasses a broad spectrum of entities. It is also unclear whether the definition should include all symptoms or just one. Thus, is why it would be important to show how many cases were defined as case definition 1 o r case definition 2 and then which ones met the inclusion criteria. to modify the figure 1. In case you do not have or can not show the suggested data, explain better the screening and recruitment method. Were all patients who met a case definition (1 or 2) assessed to evaluate inclusion and exclusion criteria, and then it was defined that they entered the study?

Author response:

We revised the methodology to explain the two-step screening and diagnostic process. General physicians screened suspected cases using broad criteria (either otoscopic or systemic). All children who met these criteria were referred to an ENT specialist who confirmed AOM diagnosis using AAP standards. Only confirmed cases were assessed for eligibility. Unfortunately, the number of cases that met definition 1 or 2 separately was not recorded systematically. Figure 1 was not modified, but the recruitment explanation was clarified in the methods section.

Comment 6

Reviewer comment:

Regarding MEF sample collection, no references are cited to support the procedure used in this study. Is the sampling protocol proprietary? Was it adapted? I request the authors to provide further information in this regard.:

Author response:

We have now cited published methods to support the sample collection protocol (references 34 and 35).

34 Fekete S, Juhász J, Makra N, Dunai ZA, Kristóf K, Ostorházi E, et al. Characterization of middle ear microbiome in otitis media with effusion in Hungarian children: Alloiococcus otitidis may potentially hamper the microbial diversity. Heliyon [Internet]. 15 de noviembre de 2024;10(21):e39380. Disponible en: https://www.sciencedirect.com/science/article/pii/S2405844024154114

35. Marchisio P, Esposito S, Picca M, Baggi E, Terranova L, Orenti A, et al. Prospective evaluation of the aetiology of acute otitis media with spontaneous tympanic membrane perforation. Clin Microbiol Infect [Internet]. 1 de julio de 2017 [citado 5 de mayo de 2025];23(7):486.e1-486.e6. Disponible en: https://doi.org/10.1016/j.cmi.2017.01.010

Comment 7

Reviewer comment:

Since the authors' results suggest existing differences in the groups' characteristics, it would be worthwhile to analyze the difference through the appropriate statistical tests and add the p-value of this result to Table 1. This would evidence the differences.

Author response:

Thank you for your valuable observation. We agree that it would have been relevant to apply inferential statistics to compare the groups with respect to their sociodemographic characteristics. However, due to the small number of patients with S. pneumoniae (n = 6), the sample lacks sufficient statistical power to support meaningful comparisons. Therefore, we chose to present the data descriptively to avoid overinterpretation of underpowered analyses. Consequently, we also modified redaction in Results, in terms of not suggesting “differences” among groups: Line 241: “Among patients with AOM caused by microorganisms other than S. pneumoniae, thirty-three (60%) were male, with a median age of 12 months (IQR = 7 - 24).”

Comment 8

Reviewer comment:

There appears to be a difference in median age between patients with PNS and those with other causes, which should be discussed further. Could this be related to the increase in transient SPN colonization of the nasopharynx in school children?

Author response:

While children with S. pneumoniae were older, we cannot confirm statistical significance due to small sample size. We did run a Mann Whitney Test comparing age between the two groups and there was not statistical difference, however, we recognize the lack of power to ensure this result is not spurious. Therefore, we consider it not appropriate to make inferences about these differences in age.

Comment 9

Reviewer comment:

In the discussion, when referring to the vaccine recommendation, other types of vaccines, such as PCV10-SII (PNEUMOSIL) that protect against SPN serotypes 1, 5, 6A, 6B, 7F, 9V, 14, 19A, 19F, and 23F are not discussed. PCV-15 and PCV-20 are not mentioned either. Likewise, this recommendation does not make sense because the phenomenon of capsular replacement in other countries has already shown that once the vaccine is changed, the circulating serotypes begin to change in the following 5-10 years. Colombia started vaccination with PCV-13 3 years ago after many countries included PCV-15 in their programs. The discussion about the recommendation for the next vaccine should revolve around vaccines after PCV-13. Suggested references PMID (38336559, 39591182, 39153492)

Author response:

We have restructured discussion in regard to vaccine recommendations. We need, however, to recognize that PCV-13 is the current officially implemented vaccine in Colombia. Therefore, we framed our interpretation within this context.

In regard to other vaccines, we added a paragraph stating:

“As vaccine coverage expands to more serotypes, the replacement effect by non-vaccine serotypes (41) or other microorganisms will continue, as will the antimicrobial pressure due to the indiscriminate use of antibiotics. Among the cases of non-pneumococcal AOM, it was found that 25.5% of the participants received the PCV-13 vaccination schedule. This observation could serve as a baseline for future follow-ups to assess the effect of replacement by other microorganisms and serotypes. Pneumococcal surveillance to non-invasive disease, including serotyping and resistance profiling, is critical for tracking the community-level circulation of multidrug-resistant strains. We acknowledge the development and licensure of newer pneumococcal conjugate vaccines (PCV10-SII, PCV-15 and PCV-20) which offer broader serotype coverage and are under evaluation or currently used in various immunization programs of other countries (49–51), but these vaccines were not yet licensed in Colombia at the time of this study.”

Comment 10

Reviewer comment:

Although one of their important findings is that serotype 19a was found in a higher proportion, they do not discuss how, in Colombia, this circulating serotype is also the major cause of IPD PMID (34641809,32964223).

Author response:

Serotype 19A has emerged as a predominant serotype also in IPD. Its multidrug resistant pattern could be associated also to worse clinical outcomes such as IPD.

We added a sentence to highlight this: “Evidence of a similar behavior in Colombia regarding isolates of S. pneumoniae causing IPD was found, wherein 19A and 6C are predominant serotypes.”

Comment 11

Reviewer comment:

There are paragraphs in the discussion that merely mention facts but do not contradict or support the researchers' findings and do not contribute to the scope referred to in the objective. Examples of these are found in lines 344-347, 357-367, and 386-389.

Author response:

Thanks for your detailed revision of Discussion and meaningful suggestions. We revised the discussion and removed or integrated the identified sections into more cohesive and relevant paragraphs.

Comment 12

Reviewer comment:

I kindly suggest refining the information in the discussion to make the reader understand in a more adequate way the importance of the findings and how this study is the basis for further research on pneumococcal diseases different from IPD, as well as how the impact of the vaccination that has been applied to the Colombian context is reflected in its results and could be modified with the introduction of other vaccines.

Author response:

We reformulated the discussion to highlight key contributions: comparison with national studies, impact of vaccination on serotype distribution, the emergence of serotype 19A, and clinical aspects such as seasonality. Paragraphs were condensed to improve focus and readability.

Sincerely,

Nelson Alvis Guzman. PhD

Corresponding author

Universidad de la Costa

nalvis@cuc.edu.co

---

## [Decision Letter · Decision Letter 2]

Evaluation of Streptococcus pneumoniae as a cause of acute otitis media in Colombia: a prospective study

PONE-D-24-41495R2

Dear Dr. Alvis Guzman,

We’re pleased to inform you that your manuscript has been judged scientifically suitable for publication and will be formally accepted for publication once it meets all outstanding technical requirements.

Kind regards,

Luis Felipe Reyes, M.D., Ph.D., MSc.

Academic Editor

PLOS ONE

Reviewers' comments:

Reviewer's Responses to Questions

**Comments to the Author**

Reviewer #1: All comments have been addressed

Reviewer #3: All comments have been addressed

2. Is the manuscript technically sound, and do the data support the conclusions?

Reviewer #1: Yes

Reviewer #3: Yes

3. Has the statistical analysis been performed appropriately and rigorously?

Reviewer #1: Yes

Reviewer #3: Yes

4. Have the authors made all data underlying the findings in their manuscript fully available?

Reviewer #1: Yes

Reviewer #3: Yes

5. Is the manuscript presented in an intelligible fashion and written in standard English?

Reviewer #1: Yes

Reviewer #3: Yes

Reviewer #1: Thank you for addressing all the comments so thoroughly. I believe the manuscript is now in much better shape and is clearly grounded in the significance of the findings. In my opinion, this is a strong manuscript, and I recommend it for publication.

Reviewer #3: The authors have addressed all the comments raised adequately. The language is grammatically correct and conveys the meaning to the reader.

**Do you want your identity to be public for this peer review?** For information about this choice, including consent withdrawal, please see our Privacy Policy

Reviewer #1: No

Reviewer #3: No

---

## [Editor Report · Acceptance letter]

PONE-D-24-41495R2

PLOS ONE

Dear Dr. Alvis Guzman,

I'm pleased to inform you that your manuscript has been deemed suitable for publication in PLOS ONE. Congratulations! Your manuscript is now being handed over to our production team.

Kind regards,

on behalf of

Dr. Luis Felipe Reyes

Academic Editor

PLOS ONE